# The Biological and Molecular Action of Ozone and Its Derivatives: State-of-the-Art, Enhanced Scenarios, and Quality Insights

**DOI:** 10.3390/ijms24108465

**Published:** 2023-05-09

**Authors:** Valter Travagli, Eugenio Luigi Iorio

**Affiliations:** 1Dipartimento di Biotecnologie, Chimica e Farmacia, Università degli Studi di Siena, Viale Aldo Moro 2, 53100 Siena, Italy; 2International Observatory of Oxidative Stress, 84127 Salerno, Italy; 3Campus Uberlândia, Universidade de Uberaba (UNIUBE), Uberlândia 38055-500, Brazil

**Keywords:** ozone biochemistry, ozone treatment, molecular action, ozone derivatives, quality issues

## Abstract

The ultimate objective of this review is to encourage a multi-disciplinary and integrated methodological approach that, starting from the recognition of some current uncertainties, helps to deepen the molecular bases of ozone treatment effects on human and animal well-being and to optimize their performance in terms of reproducibility of results, quality, and safety. In fact, the common therapeutic treatments are normally documented by healthcare professionals’ prescriptions. The same applies to medicinal gases (whose uses are based on their pharmacological effects) that are intended for patients for treatment, diagnostic, or preventive purposes and that have been produced and inspected in accordance with good manufacturing practices and pharmacopoeia monographs. On the contrary, it is the responsibility of healthcare professionals, who thoughtfully choose to use ozone as a medicinal product, to achieve the following objectives: (i) to understand the molecular basis of the mechanism of action; (ii) to adjust the treatment according to the clinical responses obtained in accordance with the principles of precision medicine and personalized therapy; (iii) to ensure all quality standards.

## 1. Introduction

The term “ozone therapy” is easy to understand as it puts in a nutshell the therapeutic treatment with ozone and its derivatives. However, this apparent intuitive simplicity has its pitfalls. In fact, ozone and its derivatives have a multitude of healing uses, each characterized by specific biological and molecular aspects that must be considered to have ozone treatments of the highest possible quality.

Regarding the history of ozone from the point of view of its discovery and its chemical–physical properties, please refer directly to a series of exhaustive articles on the subject by Rubin [1,2,3,4,5,6,7,8].

Specific to aspects of general and inorganic chemistry, another milestone is Horváth’s book. In this text, the physiological effects of ozone begin to be dealt with, albeit related to the reaction of ozone with the mucous membranes of the upper respiratory tract [9].

The context is different regarding the clinical use of ozone and its derivatives. First, it should be pointed out that, in recent years, some reviews that often contain gross mistakes have been published in international journals of unknown scientific authority. What is more, some have several citations currently exceeding 200 according to Scopus, despite the need to correct what they report [10].

It is becoming increasingly important to make researchers aware of the full understanding of the various aspects related to the therapeutic use of ozone so that implicit quality standards are met.

## 2. Biological and Molecular Action of Ozone

The actions of atmospheric ozone have been well known since as early as the 1950s, when the harmful effects of high ozone levels on humans were first recognized in Los Angeles [11]. Similar observations had previously been considered in the plant world, again around Los Angeles, where high ozone levels were first connected with spots on the leaves of garden plants and crops and subsequently with the increased mortality of ponderosa and Jeffrey pines [12].

The biological effects of ozone on man and animals have subsequently been studied, where the ozone interactions with lung tissue represent the biochemical basis of ozone toxicity [13,14]. Ozone ability to cause oxidation or peroxidation of biomolecules directly and/or via free radical reactions has been and continues to be the basis for its wide fields of applications as an agent for decontamination preservation and storage, disinfection, and removal of odors [15,16].

In justification of the fact that toxicity or therapeutic application are not absolute concepts but must be contextualized, the reader is referred to consider what can happen with erroneous intakes of water that lead to true “water intoxication” [17]. On the other hand, the same properties for a superficial interpretation may justify ostracism towards the therapeutic use of ozone. On the contrary, the holistic view of each organism represents the basis for the therapeutic application of ozone and its derivatives from a personalized medicine perspective in the view of an oxidative eustress concept [18,19].

However, before deciding on the time scale of any potential medical use of ozone, it would appear helpful to deal with both the kinetic aspect of ozone oxidation and the typical ozone concentration in liquids, mainly aqueous. Many factors have relevant effects on these aspects. In fact, the effects of pH, salinity, concentration of organic carbon, temperature, and hydrodynamic conditions have been described in several discordant ways by many researchers, with a special finalization for industrial applications [20,21,22]. In view of the complexity of the biological systems with which ozone comes into contact, the definition of general relationships between ozone and its initiators, promoters, and scavengers (especially the effects of organic matter) to obtain a univocal model that can be used in practical applications remains very unlikely. Moreover, a detailed investigation of molecular effects of ozone oxidation on amino acids and proteins and their subsequent fates could contribute information of interest [23]. On the other hand, it should be emphasized at the outset that the ranges of ozone concentration usually employed during therapeutic treatments do not go so far as to significantly affect these components.

## 3. Clinical Action of Ozone

The features to be taken into consideration in the case of the use of oxygen–ozone gas mixtures for therapeutic purposes are many. It should be reminded that ozone in these conditions never exceeds 5% of the gaseous mixture used. The maximum ozone concentrations in this regard can be approximated to 100 mg/L in medicinal oxygen as the carrier. The importance of dosage justifies the fact that toxicity or therapeutic applications are not absolute concepts but must be contextualized, as previously stated [17]. According to this approach, ozone sees its rationale for use correlated with the route of administration, quantity, and mode adopted [24,25]. However, for a correct understanding of the clinical properties of ozone, first it is appropriate to provide some clarity on the following aspects concerning the correct therapeutic framework of ozone treatment.

Ozone treatment is not an alternative medicine.

Ozone treatment is not an alternative medicine. Such a statement is possible mainly thanks to clinical research and scientific studies by Prof. Bocci, overall represented by his books on the subject [26,27,28]. Ozone treatment is based on the general principles that have made Western medicine remarkable: strong biochemical bases, scientific analysis of metabolism in all its complexity, research into the real causes of diseases, and use of all the diagnostic and therapeutic tools that scientific progress provides. It is compatible with conventional medical therapies. In fact, apart from the fragmentation of modern medical specializations, each patient, donor, and product application is unique; therefore, the field faces complexities in the development of new products and therapies that are safe and effective that are not faced by developers of more conventional therapies. Because clinical needs are diverse, many different treatment approaches and models may be used; this will necessitate a variety of manufacturing pathways and standards to ensure safety and efficacy. The patient’s condition, the physiological context of the treatment, and the delivery of the therapy are all variables that factor into the clinical application of a regenerative engineering therapy. By considering these elements, along with the critical quality attributes of a product in the early stages of development and manufacturing, it may be possible to improve patient outcomes by increasing product consistency. Collecting more data could lead to a better understanding of these variables and how to plan and account for them in the early stages of product research and development.

Ozone is not a pro-drug.

Ozone is not a pro-drug in the proper sense of this term. In fact, a pro-drug is a pharmacological inactive derivative of a drug molecule that can release the parent molecule quantitatively due to an enzymatic or spontaneous reaction in the body. The chemical group used for derivatization of the parent drug molecule, called the pro-group, should be non-toxic. The pro-drug term involves a chemically modified inert compound that, upon administration, releases the active parent drug to elicit its pharmacological response within the body [29]. Ultimately, a pro-drug is a biologically inactive molecule that, once introduced into the body, undergoes chemical—mainly enzymatic—transformations that enable it to be converted into a drug. Typically, a pro-drug improves certain characteristics, such as transport, absorption, and diffusion in the body, and is divided into two categories:
○Carrier pro-drug: where the drug is bound to a molecule that has a transport function.○Bio-precursor: the molecule acts as a substrate to enzymes, giving the active drug.

Moreover, from a molecular point of view and in relation to the absence of specific receptors and the extreme reactivity with the biological liquids with which it comes into contact, gaseous ozone cannot be considered a drug in the common sense of this term [30]. On the contrary, the contact of ozone with biological matrices results in the release of effector molecules to tissues because of its chemical reactivity. As a rule, these are molecules of low molecular weight that selectively bind to a protein, regulating its biological activity. Effector molecules are both hydrophilic and lipophilic species acting as ligands that can increase or decrease enzyme activity, gene expression, or cellular signals [31,32]. Thus, a great number of cells in various organs upregulate the synthesis of anti-oxidants that are significantly able to counteract the excess of reactive oxygen species (ROS) that are extremely deleterious in metabolic diseases where a chronic oxidative stress is present [33,34,35].

Once these aspects were clarified, some of the earliest clinical applications of ozone through inhalation were described in 1870 by Lender, a German physician [36]. At that time, Pagel critically wrote about Lender’s exaggerations: “According to Lender, ozone is a true panacea. The one-sidedness of the author was to blame for this versatility of his remedy” [37].

Apart from this pioneering phase involving the ineligible inhalation route of ozone by analogy with the knowledge of the time concerning oxygen, if by ozone treatment we mean any therapeutic use of ozone, one of the earliest applications can be traced back to 1892, when a series of case reports was published on the oral intake of both ozonated water and ozonated oil in 15 patients with pulmonary tuberculosis, as reported in an annotation published in the prestigious journal *The Lancet* [38].

Thus, a broad meaning of the term “ozone treatment” identifies any therapeutic modality carried out using ozone and its derivatives, as shown in Table 1.

These modes of administration were empirically taken into account, at least until the 1990s, i.e., before the molecular and biochemical basis of ozone’s therapeutic action was more scientifically addressed by Prof. Bocci, as mentioned above. Such categories do not refer to dental and veterinary applications, for which please see recent specific reviews [39,40]. Modes of administration using infiltrative therapy may also occur where a different modality is generally applied [41]. Similarly, the trans-cutaneous route has not been mentioned, despite Prof. Bocci’s pioneering remarks about the oxidant effect of ozone in several tissues and body fluids [42].

As illustratively shown in Figure 1, the considerations to keep in mind for a correct interpretation of ozone treatment change a lot depending on the route and method of administration. In detail, drawings exemplify: (i) the considerable qualitative and quantitative variability of the chemical species that can be formed when the gaseous ozone impacts a biological matrix. In a nutshell, the two categories of compounds that originate can be indicated as reactive oxygen species (ROS) and lipid oxidation products (LOP). Both ROS and LOP represent the effector molecules responsible for modulating the therapeutic activity in the organism. From a kinetic point of view, the former are the first to develop [31]. The species and the quantities that are formed (Figure 1, top, colored spheres) obviously depend on both the anti-oxidant background of the substrate and the number and type of reactive molecules. In fact, the same living body can give rise to different reaction products according to the contextual health conditions. By extending this concept to all the biological matrices with which the ozone comes into contact, schematically, the interlocking geometric shapes can be considered a criterion of representation (Figure 1, center, puzzle pieces). Moreover, at the level of systemic administration, it must be considered that oxygen is also re-infused in a venous environment (Figure 1, bottom, different red–blue nuances). The actual role of free oxygen content at the level of the vascular context, and generally at the administration site, is still to be fully studied [43,44,45].

Obviously, the theoretical considerations indicated with regard to either locoregional or systemic activity must consider the possible deviations achievable in medical practice. Critical issues can be related to the lesion type as well as to the procedure modalities. From this point of view, there are considerable efforts required for a standardization of concentration, volume, site, needle, injection time, and contact time, just to name a few. After all, there is increasing awareness of the inter- and intra-individual variability of treatment outcome. In this regard, the “imprecision medicine” is intrinsic to the complexity of human biology systems, even in the field of the current personalized medicine model [46,47].

Recently, to complicate the picture of the ozone systemic administration [48], a discussion was opened about the 10-passes ozone high-doses therapy (OHT) as a safe and effective therapeutic option for pharmacological therapy in a variety of chronic and acute diseases. In fact, although this modality has the merit of being able to guarantee a certain standardization, it finds a certain discordance in terms of the hormetic approach, now recognized in systemic ozone treatment [49,50].

## 4. Potential Effects of the Interactions between Ozone and Its Putative Molecular Targets

In the context of systemic ozone treatment, it now seems evident that ozone acts by inducing, directly or indirectly, oxidative changes in cellular/acellular components of our organism. It is beyond the aim of this paper to analyze such aspects as they have already been well summarized in brilliant articles previously cited. Therefore, the goal is not to summarize or update what many researchers now consider certainties but rather to try to shed light on the other or dark side of the moon, i.e., the less known one, related to some potential effects of the interactions so far not yet taken sufficiently into consideration, between ozone and its putative molecular targets. Of course, the kinetic aspects of the interaction between ozone and its potential molecular targets, and, therefore, the oxidation kinetics of ozone with blood, remain to be determined. It is a methodologically complex work given the heterogeneous nature of the biological matrix in question, which can be traced back to a colloid in which “living” non-inert cellular components are dynamically included, capable of influencing in an unpredictable way the response to ozone and, therefore, the kinetics of oxidation. It is hoped that, in the future, the development of in vitro models, such as those successfully applied, for example, to wastewater, will provide valuable information useful for better understanding the molecular basis of ozone therapeutic treatments [51]. Following a mechanistic and probabilistic approach, and assuming that ozone acts through direct and/or indirect oxidizing mechanisms, the biological and clinical effects of ozone treatment could be due to the electrophilic attack of specific molecular targets located inside or outside the cells. As shown in Figure 2, these elective targets are—primarily and most probably—the double bonds between carbon atoms and free thiol groups, belonging to two large molecular networks, respectively, of unsaturated fatty acids/sterols and of peptides/proteins, showing at least one pair of potentially interacting residues of cysteines. Other targets may include classically redox-sensitive chemical species, showing either reducing (e.g., albumin) or oxidant (e.g., nitric oxide) activity, and many other compounds [25,32,52,53,54,55].

Of course, some such effects can be hypothesized/proven during the ex vivo ozonation process, while others are hypothesized/proven mostly after blood re-infusion. When indicated, however, we will report and discuss even data from ozone toxicity studies. Such effects must be always contextualized, taking into account the dosing and the intervals of treatment/exposure due to the documented preconditioning effect of ozone, and so on.

With this premise, the oxidative attack carried out on the double bonds can lead to two different consequences, i.e., lipid peroxidation and cis–trans isomeric conversion.

Lipid peroxidation generates lipo-peroxides as its main intermediates [56]. Lipo-peroxides, if exposed to free transition metals (e.g., iron or copper), can undergo the Fenton’s reaction, thus generating highly reactive oxygen-centered alkoxyl and peroxyl radicals [57]. After acting, lipo-peroxides are inactivated to organic alcohol derivatives by glutathione peroxidase (GPx) [58]. At supra-physiological levels, if their clearance (by GPx deficiency) is impaired, in the presence of enough amount of available oxygen and decreased anti-oxidant power, lipo-peroxides, in addition to generating increased amounts of highly reactive oxygen radicals (if a chelation deficit occurs), can be further oxidized and fragmented to malondialdehyde, i.e., a suitable plasma bio-marker of oxidative distress [59]. The final effect can be different if the molecular target is free or included in another bigger molecule.

Evidence shows that ozone exposure, in different experimental and clinical models, is associated with lipid peroxidation, which can be detected with highly specific analytic techniques [60,61,62,63,64,65,66]. Therefore, by oxidatively modifying the double bonds of isolated molecular targets, ozone treatment can virtually interfere with the functions controlled by: (i) lipid peroxides, including cell signaling and ferroptosis, a recently discovered mechanism of cell death [67,68,69,70,71]; (ii) the endocannabinoid system, i.e., a widespread neuro-modulatory network involved in the developing central nervous system and in tuning many cognitive and physiological processes, including pain control and immunity [72,73]. Indeed, among endocannabinoids, synaptamide is the ethanolamide of docosahexaenoic acid (i.e., a poly-unsaturated essential fatty acid belonging to the omega-3 family), while anandamide is the ethanolamide of arachidonic acid (i.e., a poly-unsaturated essential fatty acid belonging to the omega-6 family) [66,72,73]. Therefore, it can be argued that some favorable neuropsychic-antalgic-immunomodulatory effects of ozone can be virtually related to its interactions with the endocannabinoid system. On the other hand, the oxidation of cholesterol and its derivatives can act either as a pro-apoptotic signal or as a pro-inflammatory challenge [74,75,76]. In this subject, it could be extremely interesting to evaluate whether the anti-inflammatory effect ascribed to ozone treatment is attributable to the activation of the specialized pro-resolving mediator network, related, as endocannabinoids, to the metabolism of poly-unsaturated essential fatty acids [77]. Promisingly, the exposition to ozone of female C57BL/6J mice was associated with an increase of specialized pro-resolving mediators, such as resolvin D5, in lung tissue [78]. In the same animal model, lung inflammation following ozone exposure was associated with decreased levels of resolvins [79].

Ozone treatment could also modify oxidative molecular targets inserted in phospholipids, which are components, in turn, of circulating lipo-proteins and cell membranes. In such super-molecular complexes, (poly)unsaturated fatty acids create a unique matrix, whose physicochemical integrity helps to ensure the functions of the incorporated proteins [80]. Therefore, any oxidative changes of such (poly)unsaturated fatty acids may impact both lipo-proteins and cell membranes with unpredictable outcomes.

In blood plasma, ozone could act not only on free (poly)unsaturated fatty acids (bound to albumin) [25] but also on esterified ones (which represent the majority) associated with the various families of circulating lipo-proteins. Lipo-proteins, in fact, carry at least two enzymes, i.e., the LDL-associated paraoxonase-1 (showing antioxidant and detoxifying activity) [81] and the HDL-associated PAF-acetylhydrolase (showing anti-inflammatory activity) [82]. Moreover, lipo-proteins also transport some lipophilic anti-oxidants [83], such as ubiquinol [84] and, depending on the diet, tocopherols and carotenoids, such as astaxanthin [85], whose role is to protect both unsaturated fatty acids and cholesterol from oxidation [86]. In case of a deficiency in the anti-oxidant systems, the double bonds of such lipids become virtually exposed to the oxidative attack of the ozone. One of the consequences, in addition to the impact on an inflammatory/anti-oxidant/detoxifying response, can be the conversion of native to oxidized low-density lipo-proteins (ox-LDLs), whose plasma level is considered a reliable bio-marker of oxidative distress, closely related to the evolution of atherosclerosis [87]. Preliminary evidence indicates that ozone, ex vivo, is able to oxidize low-density lipo-proteins [88]. Of course, the potential health risk of these phenomena, possibly induced by ozone treatment, remains to be demonstrated, also because ox-LDLs could function as hormetins and could possibly train and, therefore, strengthen the immune system (immune training) [89,90].

The effects of ozone on the classical blood elements (erythrocytes, leukocytes, and platelets) have been well documented [25]. However, almost nothing is known about the potential effects of ozone on other circulating cells, such as endothelial cells and senescent cells, on which research has been focusing in recent years. An increase in the levels of circulating endothelial cells, showing proliferative and vasculogenic activity, has been related to inflammatory diseases and oxidative distress, such as SARS-CoV-2 infection [91,92]. Conversely, adequate control of oxidative stress slows down the senescence of their progenitors, also increasing their functional capacity [93,94]. In this scenario, the inhalation of house dust and ozone was associated with decreased levels of late endothelial progenitor cells, a probable indicator of cardio-vascular risk, due to the combination of the inflammatory potential of the house dust and the pulmonary oxidative stress induced by ozone [95]. Senescent cells are cells showing a phenotype with a stable proliferation arrest unresponsive to mitogenic stimuli; they remain viable and can shift towards a complex senescence-associated secretory phenotype [96]. Cell senescence, which can negatively impact tissue repair and regeneration but can provide also an effective anti-tumor mechanism, is one of the 15 recently recognized aging hallmarks [97] and therefore a valuable target for successful aging strategies, among which have also been included ozone treatment [54].

In any case, extensive phenomena of uncontrolled lipid peroxidation can variously alter the control functions of cell membranes on the flow of energy, metabolites, and signals [98,99]. For instance, ozone-oxidized membrane phospholipids are powerful triggers of inflammation [100]. Interestingly, ozone can potentially oxidize circulating micro-vesicles, which are important vehicles of biologically active molecules (including nucleotides and nucleic acids) [101] (another potential, often neglected, target of ozone treatment). Moreover, ozone exposure can induce the release of pro-inflammatory micro-vesicles either in lung tissues [102,103] or in the whole blood after ozonation [104]. Since the blood circulates in the vascular system, among the potential targets of ozone, following toxic exposure or ozone treatment, it is also necessary to consider the glycocalyx of the endothelium, which is extremely sensitive to oxidative and inflammatory damage [105,106,107].

The second effect that ozone can indirectly induce on the carbon–carbon double bond of an unsaturated fatty acid is the cis–trans isomeric conversion [108]. If this phenomenon occurs in the context of a membrane phospholipid, and if it is large enough, it can cause an increase in the thickness and a reduction in the fluidity of the lipid bilayer, a well-known adaptive phenomenon observed in some bacterial species due to a conformational change that makes a trans-fat acid structurally similar to a saturated fatty acid [109]. One of the main culprits of isomeric conversion is the peroxyl radical, one of the potential by-products of ozonation [14]. An increase in the levels of trans-fatty acids in the blood and tissues of exogenous derivation (junk food) is associated with metabolic and degenerative diseases, a highly unlikely event following ozone treatment [110,111]. The second potential targets of ozone, after carbon–carbon double bonds, are thiol groups of paired cysteine residues of particular proteins [112,113]. Oxidation, in this case, is generally mediated by hydrogen peroxide and can be reversible or irreversible [114]. Examples of reversible oxidations are those involving the transcription factor Nrf2, a recognized target of ozone treatment [115], and, potentially, thioredoxins, both of which are involved in cell signaling. Irreversible oxidation can lead predictably to denaturing phenomena and undesirable structural and/or functional consequences. Organic thiols can also protect from ozone-induced oxidative damage [116]. Noteworthy, thiyl radicals, potential by-products of ozone oxidation, can promote under certain conditions the cis–trans conversion of unsaturated fatty acids [117,118,119]. Therefore, the two oxidizable networks, i.e., carbon–carbon double bonds and thiols, can also interact with each other. This opens up new scenarios on the potential effects of ozone treatment.

In addition to lipids and organic thiols, ozone can potentially oxidize, albeit with different intensity, many other circulating molecular substrates [25], ultimately and potentially influencing their functions, among which are: (i) nucleotides (miRNA), (ii) proteins (complement, coagulation, proteases/anti-proteases), (iii) hormones, and (iv) amino acids. In particular, it has been shown that they can be oxidized into hydro peroxides [120]. More specifically, ozone can react with the unprotonated groups of amino acids, except for tryptophan and sulfur amino acids. Indeed, it was reported that ozone can attack the sulfur moiety of both methionine and cysteine, thus potentially interfering with thiol homeostasis. Finally, ozone can react with the amine group of aliphatic amino acids (e.g., isoleucine, leucine, and valine), thus generating nitrate or ammonia [121]. However, the real impact of such reactions in vivo after treatment of blood with ozone is not currently known.

Ozone, due to its direct or indirect oxidizing action, can also influence the levels and/or reactivity, and therefore the biological effects, of substances freely circulating in the blood described both as pro-oxidants and as reducing/anti-oxidants. Among extra-cellular potential pro-oxidant targets, nitric oxide, a nitrogen-centered gaseous radical species, is the central element of a network involving a series of metabolites (nitrates, nitrites, peroxy-nitrite, and nitro derivatives) and enzymes (nitric oxide synthetase, arginase, heme-oxygenase, etc.), whose levels, and therefore final activity on endothelial protection, neurotransmission, defense against pathogens, etc., depend on the bio-availability of oxygen [122,123,124]. This complex scenario explains the effects—often apparently contrasting—of ozone treatment on nitric oxide; in any case, the documented favorable effects on some cardio-vascular pathologies are worthy of note [125,126].

Among extra-cellular potential reducing/anti-oxidant circulating targets of ozone, such as albumin, uric acid, bilirubin, ubiquinol, lipoic acid, tocopherols, carotenoids, polyphenols, etc., please refer to previous papers [25,26,27,28]. Promising but not yet well-investigated potential molecular targets of ozone are: (i) carnosine (β-alanyl-l-histidine) showing pH-buffering, metal-ion chelation, and anti-oxidant capacity, as well as the capacity to protect against the formation of advanced glycation and lipoxidation end-products [127]; (ii) some polyamines [128].

Among intra-cellular potential reducing/anti-oxidant targets of ozone are superoxide-dismutase, catalase, glutathione peroxidase, and glutathione inside red blood cells, all suitable markers of either ozone exposure or ozone treatment [63,129,130,131,132,133].

Outside the vascular system, the action of ozone may have an unknown number of potential targets, among which fibroblasts, macrophages, the extra-cellular micro-environment, especially of tumors, and the intestinal microbiota are assuming an emerging role. The effect of ozone on fibroblasts depends on their functional state: in non-activated ones, ozone selectively stimulates the proliferation and expression of cell-surface protrusions; on both, it activates both the anti-oxidant response and the secretion of IL-6 and TGF1b [134]. The use of ozone treatment in regenerative medicine must take into account this differentiated effect; in fact, fibroblasts are extremely heterogeneous cells and, therefore, they can respond in different ways to the same stimulus [135,136]. In other words, generalizing the effects of ozone on cell targets can be misleading.

This is particularly true for macrophages, of which different phenotypes are known (M0, M1, and M2), each with specific functional profiles variously sensitive to exposure to ozone [103,137,138,139]; unfortunately, we know little about the effect of ozone treatment on the production of pro-resolving substances that, through the M1-M2 switch, could contribute to explaining the described anti-inflammatory role of macrophages [137].

The micro-environment can be defined as the set of physical–chemical variables (e.g., radiation, osmotic pressure, partial pressure of gases, metabolites, micro-vesicles, etc.) capable of acting on a discrete three-dimensional area of space in close contact with the surface of a limited number of cells and any micro-organisms. Borrowed from oncology, the micro-environment, overcoming the old concept of the extra-cellular matrix, is today considered one of the privileged interlocutors of the biological action of ozone on cells and on any viruses, bacteria, or fungi associated with them; relevant examples of the micro-environment are the discrete areas of endothelial surface exposed to blood or the smallest areas of intestinal mucosa exposed to the gut lumen, with its microbiota [140,141]. In this scenario, today we know that the tumor micro-environment, occupied by a series of micro-organisms (e.g., virus, bacteria, fungi) [142,143,144,145,146], whose role remains to be determined, can conditionate the proliferation and metastasis of neoplastic cells, which is a very promising field, given the evidence in favor of the anti-microbial and potential chemo-therapeutic role of ozone [25,147].

Finally, since humans are holobiont, any biological effect induced on one of two components (macro and micro) will influence the functionality of the other and vice versa [148,149,150]. The redox system is one of the biochemical universal interspecies links between the intestinal micro-biota (above all its bacterial component) and the cells of proper human tissues [151,152,153]. Another great challenge of ozone treatment are the interesting data from the changes in 334 metabolites induced by ozone on mice serum metabolome [154].

The theory of oxidative stress can provide a key to understanding at least some of these events, as shown again in Figure 2. In a nutshell, major auto-hemotherapy, correctly performed, especially if repeated over time, could work similar to a hormetin towards the redox system, one of the main systems involved in the universal mechanism of cellular reprogramming. For example, at therapeutic doses, ozone activates the oxidative eustress Nrf2/ARE signaling pathway. Uncontrolled exposure to ozone, on the other hand, causes a dysfunction of the redox system, rather than an alteration of the oxidant/anti-oxidant balance, leading to oxidative distress [155,156]. In this scenario, the impact of ozone therapeutic treatments with a modulating action on the redox system has intentionally not been taken into account. From a conceptual point of view, it could be useful to underline that oxidative stress and inflammation, although, at least in their sub-acute/chronic form, share the absence of a specific clinical picture and, therefore, require the dosage of specific bio-markers (e.g., dosage of malonyldialdehyde or C-reactive protein), to be “diagnosed”, are the expression of two completely distinct adaptive responses from both a biological and an evolutionary point of view [157,158,159,160]. Both oxidative eustress and classic inflammation are reactive phenomena with adaptive purposes. They are activated by common stimuli and involve a time-limited response aimed at neutralizing the stressor and thus allowing survival. However, the response of the redox system is characterized, compared with the inflammatory one, by its timeliness or precocity (the oxidizing reactive species are produced in fractions of a thousandth of a second), by the relative simplicity of its molecular mechanisms (a simple exchange of electrons, which starts from the addition of an electron to an oxygen molecule), by its ability to develop at negligible energy costs, and, finally, by the relative simplicity of the regulation mechanisms. Inflammation, on the other hand, is based on the synthesis and release of much more complex soluble mediators (eicosanoids, cytokines) produced or released in a short but not very short time, whose synthesis requires a considerable energy expenditure (e.g., only one of the sub-units of NFkB is composed of over 400 amino acids, which means that its synthesis requires at least 400 molecules of ATP); the extremely different chemical nature of the various inflammatory mediators makes the regulation mechanism extremely complex and this is also one of the causes of the failure of some anti-inflammatory drug strategies. On the other hand, a condition of oxidative distress can occur even in the absence of an inflammatory condition; an example is the oxidative distress associated with the overdose of some drugs (e.g., paracetamol). [161]. Conversely, there are currently no documented examples of inflammation without concomitant oxidative stress. Noteworthy is that oxidative stress appears earlier in the evolution of living organisms compared with inflammation; this suggests that, within the reactive processes, oxidative stress probably precedes inflammation. Therefore, an effective control of the redox system, such as that attributed to ozone therapy, could be of precious help in the prevention and treatment of inflammatory processes. In this background, it should be emphasized that ozone therapeutic treatments, albeit in different experimental models, are able to favorably modulate key signal pathways involved in the control of not only oxidative processes (e.g., activation of the expression of the transcription factor Nrf2) [155,156] but also of inflammation (e.g., repression of NFkB) [162] such as polyphenols [163]. At the moment, it seems certain that the two processes influence each other and that the least desirable event to which they can lead is their synergy, identified in the concept of oxi-inflammation/oxi-inflammaging [164]. Studies on the complex interactions between oxidative and inflammatory processes are still ongoing. Their results will help in the near future to better understand the molecular basis of ozone therapeutic treatments and to improve their cost/benefit ratio.

Of course, this is just an oversimplification and many questions have not yet been fully answered, e.g., of what amplitude/duration are the variations induced by ozone before and after blood infusion? What is their kinetics? What is the impact of the conditions of the subject (well-being/disease) on the presumed modulating action of ozone? By which factors (genetic, epigenetic, environmental) can this action be influenced?

## 5. Assessment and Monitoring of Efficacy

The effects of any therapy should be able to be measured with reliable bio-markers. This still remains a challenge in the case of ozone treatment, whose biological action seems to follow the principles of hormesis, without a clear correlation between dose and effects. Over time, various approaches have followed, in most cases based on the evaluation of bio-markers of oxidative stress, being universally accepted that ozone acts, directly or indirectly, by oxidizing a series of molecular targets [165]. Unfortunately, there is no general consensus on the analytical performance of the methods proposed for the measurement of the redox state in blood samples [166,167,168]. However, the relatively newly conceptualized redoxomics seems promising [169,170].

In a wider integrated analytical approach, already underlined by an illuminating pioneering study of metabonomics [171], techniques based on the coupling of gas chromatography/mass spectrometry, together with gene polymorphism analysis, methylome profile, transcriptomics, and so on, should be more extensively used in the future to clarify these aspects [34,172,173,174,175,176,177,178,179]. In this sense, the data obtained by analyzing biological samples of plants [180] and mice [154] by means of a metabolomic approach are extremely promising. They should be considered the golden standard in the identification of the main pathways at the center of redox cellular reprogramming induced not only by environmental pollution but also by therapeutic exposure to ozone.

One of the last frontiers of ozone treatment, opened by a study on rats, is the validation of a technique capable of tracking ozone in vivo [181] that provides, amazingly and finally, the possibility of documenting in healthy human subjects the “oxidant aura” generated by the oxidation of cutaneous squalene following environmental exposure to ozone [182,183].

## 6. Safety Issues

In the early 1980s, a survey regarding the observation of therapeutic incidents was conducted by the ArztGesellschaft für Ozontherapie e.V. among ozone therapists. A total of 644 therapists (about 25% of all those contacted) stated that they had treated 384,775 patients with ozone (a minimum number of 5,579,238 ozone applications) and had reported 336 incidents. A total of 33% of the incidents were temporally but not causally related to ozone treatment. Most of the accidents were the incorrect and improper use of ozone treatment. Intra-venous ozone treatment proved to be the most “accident-prone” type of application, mainly in terms of malpractice [184].

More recently, patient selection, sterility maintenance during the procedure, patient restriction compliance, and follow-up result in higher success rates and lower complication rates [185]. Complications, mostly minor but potentially serious, are underreported [186,187,188].

Last but not least, the concerns arising from the potential formation of toxic compounds due to the oxidation reaction of ozone with the chemical species present in ozonated-infusion solutions must be evaluated. Some studies have shown no formation of toxic hypochlorites or chlorates in ozonated-physiological saline [189]. On the other hand, the formation of the chlorate ion as the main product of ozonation has been recently reported [190,191]. Furthermore, the reactivity of ozone in saline solution has been indirectly shown by its rapid degradation [48]. For this reason, when the solution is re-infused into the patient, a constant ozone bubbling is required to ensure the respect of nominal concentration of dissolved ozone [192].

Moreover, in the intra-vascular in vivo blood treatment in such a dynamic mode, the molecular correlations with ozone and blood components are the result of instantaneous and consecutive reactions that are hard to standardize. Ultimately, the reaction between ozone and whole blood takes place directly in the vein, with possible foaming in the circulatory stream to be expected, with each drop of infusion aqueous solutions carrying dissolved ozone in a chemically reactive form. In view of the “primum non nocere” Hippocratic tradition, further investigation would be desirable in treatments with ozonated salt solutions [193]. The same precautionary approach should also be applied to other infusion solutions that may have been treated with ozone prior to infusion (e.g., Ringer lactate, dextrose 5% in water); all the more reason why the therapeutic practice of injecting the gaseous oxygen–ozone mixture directly into the vein should not find application in medical practice [194].

## 7. Extemporaneous Preparation of the Gaseous Oxygen–Ozone Mixture

For vegetable matrix derivatives obtained as a result of ozonation of olive, sunflower, and similar oils, it is appropriate to establish the product “shelf-life”. In fact, the active moiety identifiable as “ozonide” and chemically corresponding to the 1,2,4-trioxolane ring, represents a stable structure obtained by rearranging the double bonds that have undergone an ozone attack in an anhydrous environment [195]. On the other hand, a unique standard for the quantification of 1,2,4-trioxolane is under examination [196]. However, from an analytical point of view, at least for the moment, there are no consistent or reliable methods to obtain validated and reproducible values [197]. Thus, physicochemical characterization of any ozonated oil must be performed for relating the right ozonation degree that fosters the required therapeutic effects. This specificity also leads to a further consideration: mixing these chemical species with each other with matrices not subjected to ozonation means diluting the number of individual portions present without changing their individual qualitative properties.

All other treatment modalities involve the extemporaneous preparation of the gaseous oxygen–ozone mixture for subsequent administration. The high instability and consequent lack of shelf life of gaseous ozone leads to the need for extemporaneous preparation, resulting in the knowledge of principles of ozonation and its equipment [198]. Therefore, knowledge of the properties of gas mixtures that are used for therapeutic purposes, either directly or solubilized in infusion liquids, is essential from this point of view.

## 8. Standardization Criteria

Healthcare providers follow the guidelines promulgated by professional bodies of which they are members. The fundamental principles on which all guidelines are based are for healthcare providers to promote the health and well-being of patients and to practice the science and art of medicine to the best of their ability and within the limits of their expertise. These are based on the highest ethical standards of medical practice.

Therefore, if the gaseous mixture is directly inoculated into the affected districts, the good clinical practices issued by the various scientific societies operating in the field define anatomical positions, volumes, concentrations, syringes, needles, and injection time.

Other practical aspects, such as the amount of blood to be treated, the choice of anti-coagulant and its mode of use, the mode and time of contact between the blood and the gas mixture, and the re-infusion rate, are very important aspects for the standardization of the ozone administration method.

Quite rightly, the so-called major auto-hemotherapy (M-AHT) can be defined as an ex vivo extra-vascular treatment of autologous blood in static mode, with the reaction of the appropriate amount of ozone involving a defined pool of whole blood outside the body to be re-infused to the same patient in an appropriate time interval. For the sake of completeness, at least for the time being, there is both insufficient experimental evidence and rational aspects to justify methods aimed at increasing both the number of consecutive treatments and ozone dosage of the same whole blood pool before re-infusion.

With regard to dynamic systemic treatment by intra-venous administration of ozone dissolved in infusion solutions, some aspects to be taken into account for a standardization process are: (i) ozonation mode; (ii) saturation time; (iii) material compatibility; (iv) reactivity; (v) stability; (vi) ozone quantification during treatment time.

The generation of any over-pressure, with the consequent impact on the amount of soluble ozone, and other practical aspects, such as the amount of blood to be treated, the choice of anti-coagulant and its mode of use, the mode and time of contact between the blood and the gas mixture, and the re-infusion rate, are very important aspects for the standardization of the ozone administration method.

As far as rectal insufflation is concerned, a sort of dynamic intra-vascular absorption of ozone at the level of the hemorrhoidal plexus as it is or, more likely, of the effector molecules resulting from its reactivity at the level of the mucosa concerned is plausible. The disadvantages of this method are the unpredictability of effector molecules absorption because of fecal content and the irritating effect of ozone on the mucosae [199].

The importance of comparing and sharing clinical results obtained in awareness of the vastness of the term “ozone treatment” can make a difference and allow these therapeutic practices to emerge from the limbo in which they have been placed until now [200].

Obviously, an essential condition is the standardization of treatment methods and basal conditions for the qualitative and quantitative evaluation of the effector molecules obtained. The potential dependence of clinical results on both intra-individual and inter-individual variables should also take into account the circadian rhythm, the characteristics of the circulatory stream (with particular reference to the micro-circulation), variables attributable to the application modalities, and the influence of the carrier represented by medical oxygen in the different matrices and administration sites, as previously shown (Figure 1, central panel).

Aspects to be taken into consideration in the therapeutic applications of ozone can be summarized as: (i) relative instability; (ii) possible action on the materials of which the tools, equipment, and work tools used in its manipulation are made; (iii) dosages to be used in ozone treatment applications; (iv) measures and precautions to be taken before, during, and after the treatment to perform it with maximum safety, to eliminate residual traces of gas that may remain in the equipment, instruments, and environment and to avoid possible poisoning of patients, as well as staff healthcare for involuntary or accidental inhalation of the same, for all of which several biological alarm signals are described that can help reveal the occurrence of this possible inhalation intoxication.

## 9. Participation of the Nursing Staff in the Therapeutic Treatment with Ozone

Upon wishing for ozone treatment to be employed in public facilities, specialized nurses should acquire a higher level technical scientific qualification that would allow them to operate in the levels of healthcare and perform assistance, administrative, teaching, and research functions in institutions with decision-making autonomy and with in-depth knowledge in the biological, psychosocial fields, as well as practical theoretical skills in specific and very complex technical matters of the exercise of the profession, and should also be trained in the invasive procedures that are practiced in it.

Methodological and technical assistance organization corresponding to the profile of nursing education at the education levels of the specialty in medical sciences specific for ozone treatment should be foreseen. In fact, these assistants should be able to deal with independent and dependent actions, such as general and specific attention to ozone application, as well as attention before, during, and after its application.

General precautions applicable in all procedures can be summarized as:Ensuring that the material to be used is resistant to the action of ozone.Ensuring adequate ventilation of the premises.Checking that the generator does not have gas leaks.Connecting the equipment in the required time before applying the procedure in question to stabilize the generation of the O_2_/O_3_ gas mixture.

## 10. Weakness in the Therapeutic Treatment with Ozone

The potential for a good therapeutic outcome following the use of ozone and its derivatives depends on the appropriateness of dosing and the correct mode of application. A wide range of ozone treatment applications and dosing options are available, resulting in inevitable heterogeneity and a lack of specific comparisons of interest. The considerable variability in clinical outcomes is affected by the systemic and local application of ozone. However, as mentioned, unfortunately there is still widespread confusion regarding terminology and lumping the single term of “ozone therapy” as a single “class” despite apparent variations within these treatments. Furthermore, studies aimed at directly comparing the consequential effects of different applications of ozone have not yet been conducted. Any conclusions that can be drawn must keep this in mind for proper interpretation of the results, even in the presence of primary clinical homogeneity. It is speculative to judge the best method of application, dosage and course of treatment, and safety and tolerability of ozone without adequate consideration of these aspects.

Moreover, conducting specific clinical trials may lead to different results than those found in real practice. Indeed, at present, the absence of specific markers that can give reliable indications of the prior oxidative status of the individual to be treated is a weakness.

Ultimately, the main weaknesses of current clinical trials can be attributed to the lack of standardization and adequate documentation in terms of: (i) experimental design; (ii) control groups; (iii) possible impacts of environmental and technical factors that may influence the clinical results obtained.

## 11. Conclusions

In conclusion, ozone treatment represents for the physician an implementation of the therapeutic armamentarium, with increasing scientific knowledge and expertise. On the other hand, any therapeutic treatment involving the physician’s choice of the use of ozone and its derivatives must take into account the accuracy of the dosage and also, if necessary, in terms of the stability of the extemporaneous preparation, the accuracy, personalization, and correctness of the treatment in terms of effectiveness, safety, purity, and cheapness. From this point of view, the systemic ozone treatment with the greatest standardization features is the so-called major-autohemotherapy (M-AHT).

Regardless of the approval by the regulatory agencies of ozone generators for therapeutic purposes (and precisely because of the need for extemporaneous preparation of the gaseous mixture containing ozone), compliance and certification of stringent quality and safety operational guidelines are of paramount importance. The collection and handling of medical ozone must take place on the basis of a specific standard operating procedure (SOP). In addition, since ozone is a toxic inhalation product, the ozone generators must be closed systems capable of guaranteeing no loss of ozone into the environment during use.

Continuous collaboration and interactions between national and international scientific societies should have as a common goal the drafting of the shared protocols, while being aware that the ultimate choice criteria are the responsibility of the physician and depend on the particular clinical context of the individual patient.

## Figures and Tables

**Figure 1 ijms-24-08465-f001:**
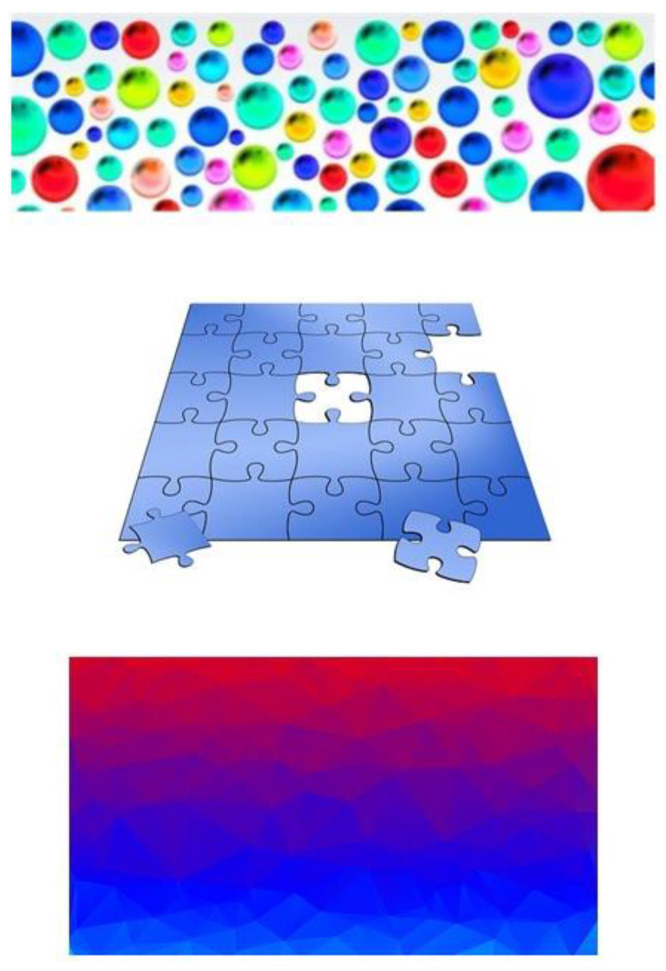
Schematic representation of the basic aspects for the therapeutic use of gaseous ozone and its derivatives. (**Top**): qualitative and quantitative variability of chemical species as effector molecules generated by the reaction of ozone with the various parts of the organism with which it comes into contact. (**Center**): specificity of effector molecules with well-defined biological aspects. (**Bottom**): autologous re-infusion of oxygenated blood in a venous environment. The actual role of free oxygen content at the level of the vascular context is under investigation.

**Figure 2 ijms-24-08465-f002:**
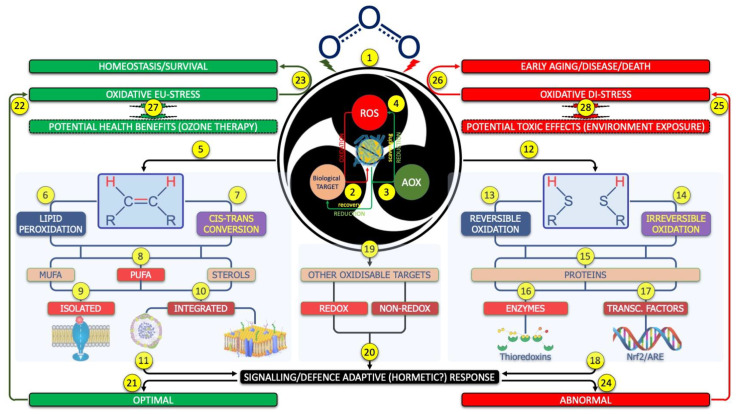
Schematic representation of the basic aspects for the therapeutic use of gaseous ozone and its derivatives. The redox system, oxidative stress, and putative biological effects of ozone. In detail, biotic or abiotic stressors, such as ozone, are believed to activate the redox system (1). The redox system is a ubiquitous evolutionary well-conserved interspecies adaptative biochemical system that exploits the exchanges of single reducing equivalent units (e.g., electrons) between a reactive oxidant species (ROS), a biological target, and a reducing/antioxidant species (AOX) to modulate signaling and defense responses. Under appropriate stimuli, ROS extracts one electron from the biological target (2), thus modulating its function; AOX returns the electron to the biological target (3), hence switching off its activity to the baseline; finally, AOX can reduce and deactivate the ROS, if exceeding, by giving it an additional electron (4). There are two main targets of ROS. As shown in the box on the left side, the first target is the double bond between carbon atoms (5), whose oxidation can lead to lipid peroxidation (6) or cis–trans conversion (7), both affecting monounsaturated fatty acids (MUFA), poly-unsaturated essential fatty acids (PUFA), and sterols, such as cholesterol and its derivatives (only peroxidation) (8). The oxidation of free fatty acids (including the PUFA moiety of endocannabinoids, such as anandamide or synaptamide) can have an impact on cell reactivity/signaling (9), while the oxidation of phospholipid-bound fatty acids could have an impact on structure and function of circulating lipo-proteins and/or cell membranes (10). The oxidation of LDL, a described possible effect of ozone ex vivo, is a mechanism involved in atherosclerosis; the lipid peroxidation and/or the isomeric cis–trans conversion can modify the thickness and/or the fluidity of cell membranes, with impairment of energetic, metabolic, and signaling flow between the two sides of them. All these changes can globally mediate the biological effects of the redox system (11). On the other hand, as shown in the box on the right, ROS can also react with the reduced thiols (12), reversibly (13) or irreversibly (14), of cysteine pairs belonging to the same protein (15), including enzymes (such as thioredoxins) (16) or transcriptional factors (such as Nrf2/ARE system) (17), thus contributing to the adaptative functions of the redox system (18). Finally, ROS can impact other targets (19) such as redox-sensitive targets (e.g., oxidants, such as nitric oxide, or antioxidants, such as bilirubin), another adaptative mechanism under the control of the redox system (20). The optimal response of the redox system (21) could lead to oxidative eustress (22), essential for homeostasis and survival (23). Unfortunately, the dysfunction of the redox system, rather than the breakdown of the balance between oxidants and anti-oxidants, can lead to an abnormal response (24) called oxidative distress (25), an emerging health risk factor related to early aging, diseases, and death (26). Regarding this background, ozone treatment should be considered as an oxidative eustress mimetic strategy (27), while the environmental/uncontrolled exposition to ozone may induce a condition of oxidative distress (28). See text for further explanation and references.

**Table 1 ijms-24-08465-t001:** Ozone Treatment Application. Main Methods.

Modality	Characteristics	Routes of Administration
Infiltration/Injection *	Amounts of gaseous mixture oxygen/ozone infiltrated/injected with a needle	Sub-cutaneous
		Peri-articular
		Intra-articular
		Per-cutaneous
		Para-vertebral
		Intra-radicular
		Intra-foraminal
		Peri-radicular
Insufflation *	Amount of gaseous mixture oxygen/ozone administered through a thin and soft polymeric catheter	Rectal
		Vaginal
		Bladder
		Auricular
Infusion *^,#^	Ex vivo static mode (major auto-hemotherapy, M-AHT)	Intra-venous
	Up to 250 mL of uncoagulated venous blood is taken into a device and bubbled with the required amount of gaseous mixture oxygen/ozone. The mixture is then immediately administered again to the same subject.	
	In vivo dynamic mode	
	Ozonation of infusion solutions, with subsequent administration of ozone solubilized in them	
	Up to 10 mL of venous blood is mixed with the amount of gaseous mixture oxygen/ozone and injected into the muscle.	Intra-muscular
Bagging *	Gaseous mixture oxygen/ozone in a bag	The relevant limb comes into direct contact with the ozone gas
Dermatological preparations ^#,†^	Ozonated water	Topical applications
	Ozonated vegetable matrices	

Legends: * oxygen/ozone, gaseous mixture; ^#^ solubilized gaseous ozone, aqueous solution; ^†^ derivatives of vegetable matrices by ozonation.

## Data Availability

Not applicable.

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
