# Peer review of "The Biological and Molecular Action of Ozone and Its Derivatives: State-of-the-Art, Enhanced Scenarios, and Quality Insights"

_ijms, 2023, doi:10.3390/ijms24108465_

Round 1

Reviewer 1 Report

The manuscript must be reviewed, because the revision does not bring anything interesting.

The discussion and conclusions are rather flat.

The discussion should be expanded to included a weaknesses paragraph.

There are grammatical errors throughout the manuscript.

Reviewer 2 Report

In this review, the authors summarize the biological and molecular action of ozone, and their effect should be appreciated. As specified by authors, the advantages of western medicine are strong biochemical bases, scientific analysis of metabolism, research into the real causes of diseases and use of all the diagnostic and therapeutic tools that scientific progress provides.  From my point of view, this review reasonably explains the biochemical bases of ozone, and I am convinced.  However, I have no comment about ozone therapy or ozone treatment. I believe that any description of ozone treatment as a potential medical use for human should remain conservative in a scientific article, unless there have been extensive recognized investigations of such treatment to human in scientific or medical literature.

Nevertheless, I have the following comments / suggestions for the biological and molecular action of ozone described in the review article:

1. while the mechanism of ozone in bio-system is well described, I would like to encourage authors to provide at least a short description about the kinetics aspect of ozone oxidation. This can be crucial for deciding the time scale of any potential medical use of ozone.

2. there should be a more detailed description of ozone oxidation of amino acids, such as tyrosine, tryptophan, histidine, cysteine, and methionine, and the subsequent fates.

3. As described by authors, both water and ozone can be toxic if  erroneous intakes (overdose) occur. While the upper limit of water intake can be easily estimated, such quantity for ozone is not trivial to be determined. For example, the actual ozone concentration dissolved in blood will depend on the Henry's law, as well as ozonated water or oil. The authors should provide a brief description based on literature, about the issue of typical ozone concentration in liquids.

4. As described in this review, the action of ozone in biological systems cause various responses similar to inflammation. Some of them may lead to eu-stress, while some of them leads to di-stress. My feeling is that this review does not deliver a systematic and sounding description about any scientific progress or analysis to distinguish and control these two pathways following the ozone oxidation, optimal and abnormal.

5. The brief introduction about the metabolism response to the products from ozone oxidation or ozone directly would be necessary.

Round 2

Reviewer 1 Report

The authors have made the requested changes. The manuscript is publishable.